# Outcome of infants with bronchopulmonary dysplasia

**Dwi Kisworo Setyowireni**[1]*, **Roni Naning**[1], **Rina Triasih**[1], **Indah Kartika Murni**[1,2]*, **Madarina Julia**[1], **Ekawaty Luthfia Haksari**[1]

1 Department of Child Health, Faculty of Medicine, Public Health and Nursing, Universitas Gadjah Mada, Dr. Sardjito Hospital, Yogyakarta, Indonesia, 2 Centre for Child Health, Pediatric Research Office, Faculty of Medicine, Public Health and Nursing, Universitas Gadjah Mada, Yogyakarta, Indonesia

* dwikisworo.setyowireni@ugm.ac.id (DKS); indah.kartika.m@ugm.ac.id (IKM)

## Abstract

### Background

Bronchopulmonary dysplasia (BDP) is the most common chronic lung disease in infants. The study aimed to evaluate the outcome of infants diagnosed with BPD in Yogyakarta, Indonesia.

### Material and methods

A retrospective cohort study was conducted by reviewing the medical records of infants with BPD born between January 2015 and December 2020 at Dr. Sardjito General Hospital in Yogyakarta, Indonesia. Surviving children were invited for clinical assessment and echocardiogram. A multivariate logistic regression analysis was used to identify predictors of mortality.

### Results

Among 8,490 newborns hospitalized in the Perinatal ward, 100 (1.2%) developed BPD, with neonatal sepsis and congenital heart disease as the most prevalent comorbidities. Of the 85 infants with complete data, 41 (48.2%) died within 7 months. Multivariate analysis revealed that post-menstrual age (PMA) < 28 weeks and mechanical ventilation as independent predictors of mortality, with adjusted odds ratios (ORs) (95% CI) of 5.27 (1.36-20.43) and 0.26 (0.09-0.75), respectively.

### Conclusions

Nearly half of the infants diagnosed with BPD died within the first seven months of life. PMA < 28 weeks was associated with increased mortality risk, while mechanical ventilation showed a protective effect against mortality in infants with BPD.

**Data availability statement:** All relevant data are within the paper and its Supporting information files.

**Funding:** The author(s) received no specific funding for this work.

**Competing interests:** The authors have declared that no competing interests exist.

## Introduction

Bronchopulmonary dysplasia (BPD) is a chronic lung disease predominantly affecting premature infants who require mechanical ventilation and oxygen therapy. However, it can also occur in term infants with severe or milder respiratory conditions. Despite improvement in perinatal care, BPD remains a significant challenge, representing the leading cause of late respiratory morbidity in preterm infants [1,2]. The long-term impact of BPD includes persistent respiratory symptoms and airflow obstruction, potentially affecting lung function into adulthood [2,3].

The incidence of BPD varies widely across institutions, reflecting differences in neonatal risk factors, care practices (such as the use of noninvasive ventilation and target oxygen saturation levels), and clinical definitions of BPD. Studies have reported BPD approximately 40% in infants born at ≤ 28 weeks gestational age [4] and up to 77% in those born at <32 weeks' gestational age and weighing <1,000 g [1]. Infants with a birth weight <1,250 g account for 97% of BPD cases [5].

Infants with BPD face considerable morbidity and mortality, with approximately 40% of those died. Survivors are at high risk for long-term complications, including airway functional impairment as exhibited by abnormal forced expiratory volume in 1 second ($FEV_1$), vital capacity (VC), and functional residual capacity (FRC) within the first five years [6]. Infants with moderate to severe BPD had lower $FEV_1$ and forced expiratory flow ($FEF_{25-75}$) than those with mild BPD, along with greater developmental delays [7], increased hospital readmission due to lower respiratory infection especially caused by respiratory syncytial virus (RSV) infections [8,9], reactive pulmonary disease, persistent airway obstruction [10], and pulmonary hypertension [11].

Despite the global burden of BPD, data on the incidence and outcomes in Indonesia are limited. Therefore, this study aimed to evaluate the outcome and predictors of mortality in children with BPD in Yogyakarta, providing valuable insights into the management and prognosis of BPD in resourse-limited settings.

## Materials and methods

### Participants

A retrospective cohort study was conducted at Dr. Sardjito General Hospital in Yogyakarta, Indonesia, from January 2015 to December 2020, involving all infants with BPD hospitalized in the neonatology ward and NICU.

### Data collection

Demographic and clinical data were collected from medical records, including date of birth, sex, gestational age, birth weight, weight-for-gestational age (Lubchenco), and pre-natal infection status. Neonatal morbidity and any re-admissions (such as neonatal sepsis, pulmonary hypertension, other pulmonary problems, congenital heart disease, and other congenital anomalies), as well as treatment and interventions (such as surfactant therapy, oxygen supplementation, and mechanical ventilation), were recorded. Mortality data, both in-hospital and post-discharge, were documented.

Surviving children were invited for clinical assessment and echocardiography between March and September 2021, with written informed consent obtained from their parents.

## Broncopulmonary dysplasia

Infants were diagnosed with BPD if they met one or more of the following criteria: (1) need for supplemental oxygen at a post-menstrual age [PMA] of 36 weeks or more [12]; (2) meeting the severity-based diagnostic criteria established by the National Institutes of Health Workshop [13], or (3) oxygen saturation below 88% within 60 minutes during a "room air challenge test" [5].

The severity of BPD was categorized based on the requirement of oxygen support, fraction of inspired oxygen ($FiO_2$), or positive pressure ventilation (PPV) at 36 weeks' gestational age, at discharge, or at 56 days old. BPD was classified as mild if infants required room air, moderate if oxygen supplementation ($FiO_2$) was < 30%, and severe if $FiO_2 \geq$ 30% or PPV was required [13].

## Pulmonary hypertension

As echo-cardiography was not routinely performed in neonates with BPD during NICU hospitalization or later follow-up. Subsequent echocardiograms were conducted for surviving infants without initial pulmonary hypertension and for those who had not previously undergone echo-cardiography.

All echo-cardiograms were performed by skilled pediatric cardiologists, with pulmonary hypertension diagnosed based on elevated pulmonary artery pressure, indicated by tricuspid valve regurgitation velocity > 3 m/s in the absence of pulmonary stenosis, or evidence of right ventricular hypertrophy with chamber dilation and a flat or leftward-deviated interventricular septum [11].

## Statistical analysis

Data were presented as mean and standard deviation (SD) for normally distributed data, median and range for non-normally distributed data, or proportions as appropriate. Survival analyses estimated median survival times, stratified by the severity of BPD. A multivariate logistic regression model identified independent predictor of mortality in infants with BPD. For this analysis, all potential predictors with p value < 0.25 in univariate analysis, were included in the multivariate model. Results were reported as odds ratios (ORs) with 95% confidence intervals (CIs), with p < 0.05 indicating statistical significance. Statistical analyses were conducted using SPSS for Mac, version 12.

## Ethical consideration

We obtained ethical approval from the Medical and Health Research Ethics Committee at the Faculty of Medicine, Public Health and Nursing of Universitas Gadjah Mada, Indonesia (KE/FK/0171/EC/2021). The study was conducted in accordance with relevant guidelines. As a retrospective study, individual patient consent was not required by the ethics committees. However, written informed consent was obtained from the parents for surviving children invited for clinical assessment and echocardiogram between March and September 2022.

## Results

During the study period, among 8,490 newborns hospitalized in the neonatology ward and NICU, only 100 infants (1.2%) were diagnosed with BPD, with complete data available for 85 of these infants. The majority of the infants were born at < 32 weeks of gestational age and had a birth weight of less than 1,500 g (Table 1). The median gestational age was 30 weeks (range: 25−42 weeks), and the median birth weight was 1,190 gram (range: 600−3,684 gram).

The incidence of respiratory problems such as cough, wheeze and dyspnea was only 8.2% among infants with BPD. On the other hand, neonatal sepsis was the most common comorbidity, especially among those with severe BPD. Infants

**Table 1. Characteristics of BPD patients.**

| | All<br>n = 85 | Mild BPD<br>n = 32 | Moderate BPD<br>n = 27 | Severe BPD<br>n = 26 |
|---|---|---|---|---|
| Gender, male (%) | 45 (52.9) | 18 (56.2) | 12 (44.4) | 15 (57.7) |
| Duration of follow up (year), median (range) | 3.67 (0.75-7.5) | 3.63 (1.08-7.5) | 4.17 (0.75-7.5) | 3.17 (1.08-7.4) |
| Gestational age (weeks), median (range)* | 30 (25-42) | 30 (25–39) | 29 (25-42) | 29.5 (26–41) |
| <32 weeks | 53 (62.3) | 20 (62.5) | 16 (593) | 17 (65.4) |
| 32–37 weeks | 23 (27.1) | 9 (28.1) | 8 (29.6) | 6 (23.1) |
| >37 weeks | 9 (10.6) | 3 (9.4) | 3 (11.1) | 3 (11.5) |
| Birth weight (g), median (range)* | 1,190 (600−3,684) | 1,300 (764−3,684) | 1,160 (600−2,597) | 1,100 (800−3,550) |
| <1000 g (%) | 20 (23.5) | 7 (21.9) | 6 (22.2) | 7 (26.9) |
| 1000 −<1500 g (%) | 43 (50.6) | 16 (50) | 14 (51.8) | 13 (50) |
| 1500 −<2500 g (%) | 16 (18.8) | 6 (18.7) | 6 (22.2) | 4 (15.4) |
| ≥2500 g (%) | 6 (7.1) | 3 (9.4) | 1 (3.7) | 2 (7.7) |
| Birth weight/gestational age** | | | | |
| SGA (%) | 21 (24.7) | 7 (21.9) | 8 (29.6) | 6 (23.1) |
| AGA (%) | 60 (70.6) | 22 (68.7) | 19 (70.4) | 19 (73.1) |
| LGA (%) | 4 (4.7) | 3 (9.4) | 0 | 1 (3.8) |
| Surfactant therapy | 16 (18.8) | 5 (15.6) | 6 (22.2) | 5 (19.2) |
| Morbidity | | | | |
| Congenital anomaly (%) | 7 (8.2) | 2 (6.2) | 4 (14.8) | 1 (3.8) |
| Prenatal infection (%) | 1 (1.2) | 1 (3.1) | 0 | 0 |
| Neonatal sepsis | 55 (64.7) | 22 (68.7) | 12 (44.4) | 21 (80.8) |
| Readmission (times), median (range) | 1 (1–3) | 1 (1–2) | 1 (1–2) | 1 (1–3) |
| Oxygen supplementation (%) | 65 (76.5) | 25 (78.1) | 20 (74.1) | 21 (80.8) |
| ≥10 days | 52 (61.2) | 20 (62.5) | 15 (55.6) | 16 (61.5) |
| <10 days | 13 (15.3) | 5 (15.6) | 5 (18.5) | 5 (19.2) |
| Mechanical ventilation (%) | 51 (60) | 23 (71.9) | 13 (48.1) | 16 (61.5) |
| ≥10 days | 38 (44.7) | 16 (50) | 8 (29.6) | 14 (53.8) |
| <10 days | 13 (15.3) | 7 (21.9) | 5 (18.5) | 2 (7.7) |
| Respiratory problems (cough, dyspnea, wheeze) | 7 (8.2) | 2 (6.2) | 3 (11.1) | 2 (7.7) |
| Mortality (all, n = 85) | 41 (48.2) | 15 (46.88) | 11 (40.74) | 15 (57.69) |
| Age (months), median (range) | 2 (1–7) | 2 (1–7) | 1 (1–6) | 2 (1–4) |
| Mortality during neonatal period (n = 41) | 19 (46.3) | 7 (46.7) | 6 (54.5) | 6 (40) |
| 1 week | 10 (52.6) | 3 (42.9) | 4 (66.7) | 3 (50) |
| 2 weeks | 5 (26.3) | 1 (14.3) | 1 (16.7) | 3 (50) |
| 3 weeks | 2 (10.5) | 2 (28.6) | 0 | 0 |
| 4 weeks | 2 (10.5) | 1 (14.3) | 1 (16.7) | 0 |

SGA-small for gestational age; AGA-appropriate for gestational age; LGA-large for gestational age; *Each sub class is compared to other sub classes; **Lubchenco.

with mild and severe BPD who received oxygen supplementation for more than 10 days were more prevalent compared to those with moderate group.

Echocardiography performed before 7 months of age was conducted in 28 infants, with 9 (32%) diagnosed with pulmonary hypertension (PH). Among those with PH, two (22%) died before reaching 7 months, compared to 7 out of 19 (37%) infants without PH (Fig 1). Congenital heart disease was detected in 35 infants (83.3%), with 74.2% diagnosed before 7

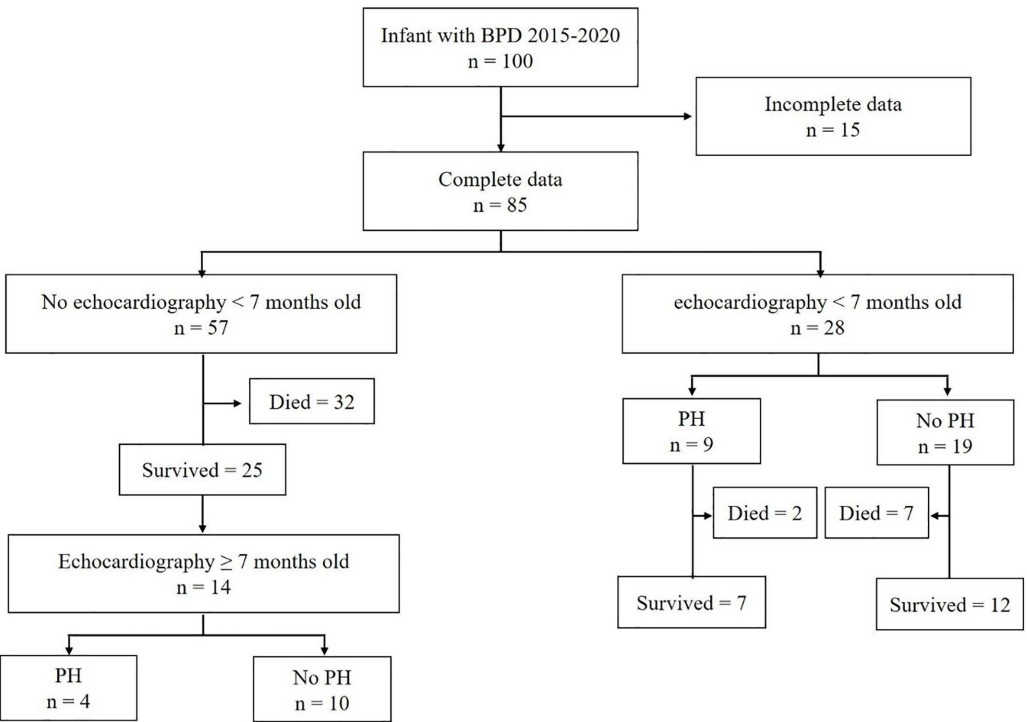

**Fig 1. Patient selection algorithm.**

months old (26/35) (Fig 1). The incidence of PH and congenital heart disease was similar across the three BPD severity groups (Table 1). Among the 14 surviving children, four (29%) were diagnosed with PH at the last follow-up (Fig 1).

Almost half (48.2%) of the infants died before the age of 7 months (Fig 2).

Table 2 shows predictors of mortality before the age of 7 months.

Univariate logistic regression revealed that PMA of < 28 weeks significantly increased the risk of mortality (OR 3.67; 95% CI 1.06–12.6; $p = 0.04$), while mechanical ventilation was associated with a decreased risk of mortality (OR 0.37; CI95% 0.15–0.91; $p = 0.03$). In multivariate analysis, PMA<28 weeks remained independently associated with increased risk of mortality (OR 5.27; CI95% 1.36–20.43; $p = 0.16$), and mechanical ventilation remained associated with a decreased risk of mortality (OR 0.26; CI95% 0.09–0.75; $p = 0.013$). However, the variable of appropriate gestational age (AGA) and congenital heart disease did not demonstrate significant associations with mortality in infants with BPD (Table 3).

## Discussion

This study evaluated outcomes and explored predictors of mortality among infants with BPD in Yogyakarta, Indonesia. The results indicated significant mortality, with independent predictors of mortality being a post-menstrual age of <28 weeks and the use of mechanical ventilation.

The incidence of BPD varies globally, with our study revealing a relatively low incidence of 1.2% among all infants and 4.8% among those born at < 32 weeks gestation. Nearly half of the infants diagnosed with BPD in our cohort died, while 32% developed pulmonary hypertension later in life. In contrast, some studies reported much higher proportion of BPD, with 49% among infants<32 weeks' gestation in Korea [14], and 34% among < 28 weeks' gestation in USA [15]. A study involving premature infants (< 37 weeks gestation) in Quebec found a BPD incidence of 36.1% [16], with 20.5% [17] classified as moderate to severe BPD. Another study among extremely low birth weight infant (≤ 1,000 g) in Birmingham

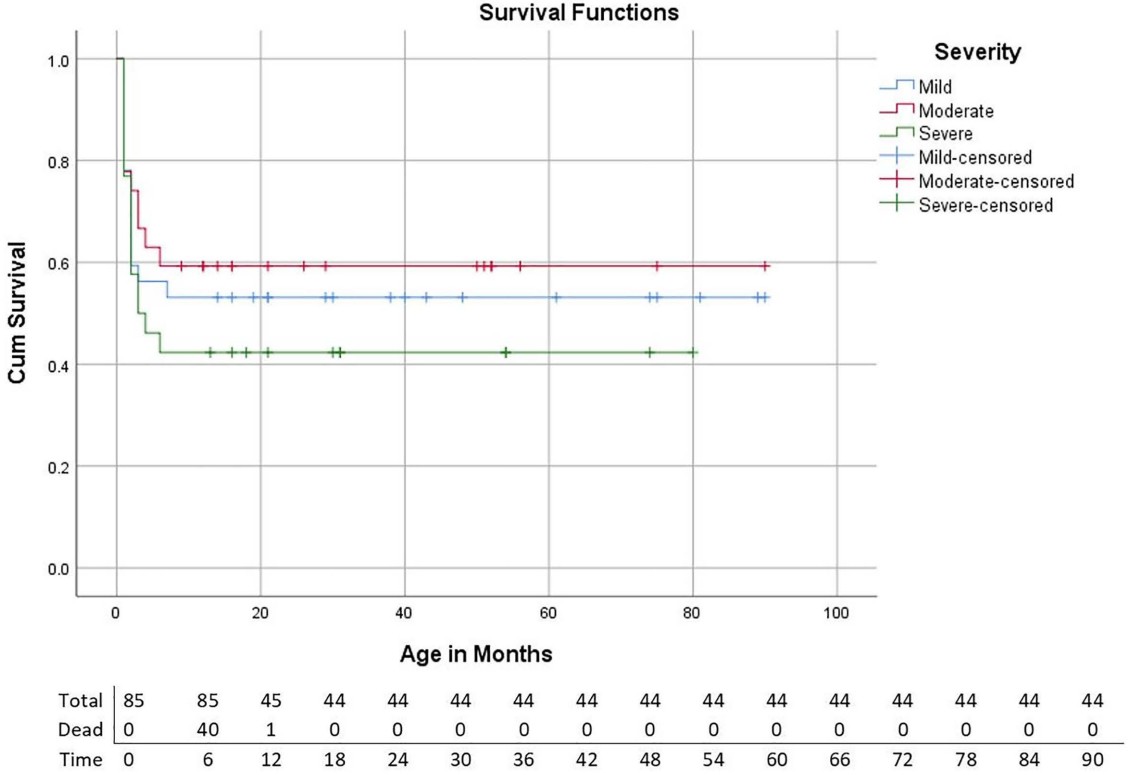

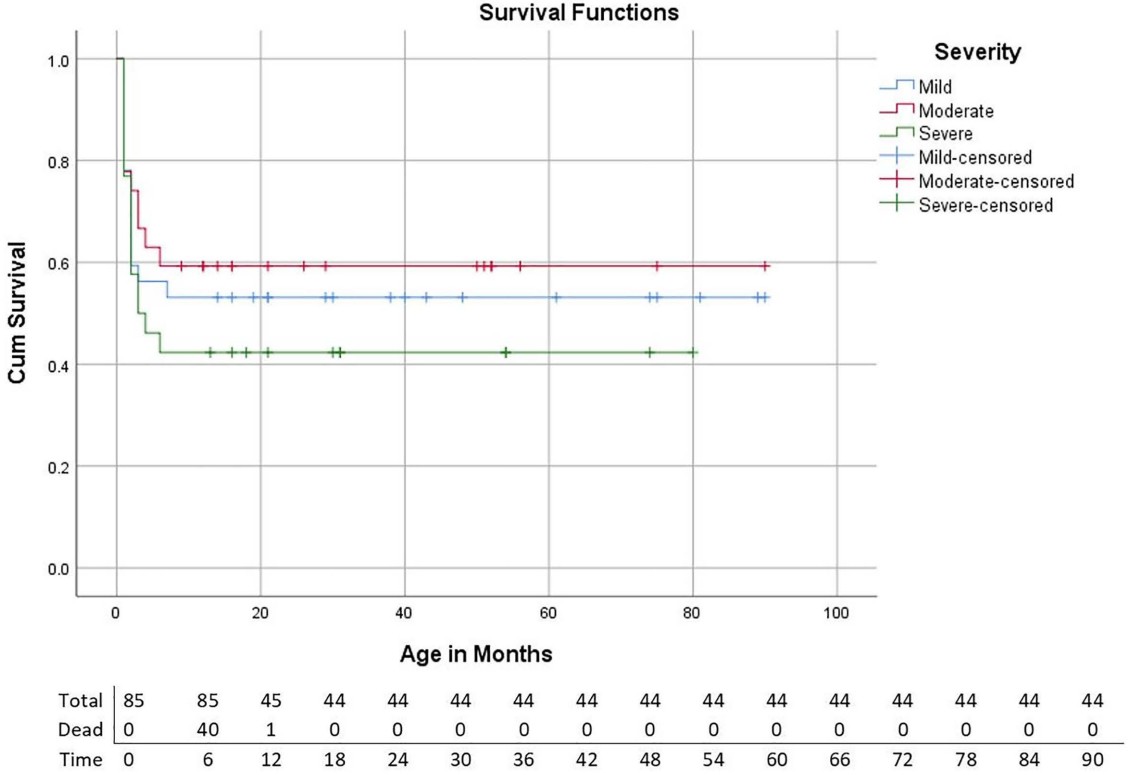

| Total | 85 | 85 | 45 | 44 | 44 | 44 | 44 | 44 | 44 | 44 | 44 | 44 | 44 | 44 | 44 | 44 |
|---|---|---|---|---|---|---|---|---|---|---|---|---|---|---|---|---|
| Dead | 0 | 40 | 1 | 0 | 0 | 0 | 0 | 0 | 0 | 0 | 0 | 0 | 0 | 0 | 0 | 0 |
| Time | 0 | 6 | 12 | 18 | 24 | 30 | 36 | 42 | 48 | 54 | 60 | 66 | 72 | 78 | 84 | 90 |

**Fig 2. Kaplan-Meier curves showing survival rate by severity of BPD infant.**

documented the proportion of moderate-severe BPD was 37.2%, and in Cincinnati was 29.4% [18]. The different findings among studies may be due to the variations in diagnostic criteria and clinical practices across different settings and time periods.

Historically, defining BPD has been a challenge, with criteria evolving over time. Early definitions primarily relied on the need for supplemental oxygen as a marker of pulmonary damage [19,20], while newer criteria incorporate clinical and radiographic changes alongside continued oxygen dependency during the first 28 days [19]. Understanding the concepts of old and new BPD may provide insights for refining diagnostic approaches and optimizing care strategies for neonates in our hospital.

The lung damage in these infants was primarily attributed to aggressive mechanical ventilation and high inspired oxygen concentrations. Infants with severe respiratory symptoms and characteristic radiographic changes presented little diagnostic difficulty. However, with the widespread use of antenatal corticosteroids, postnatal surfactant therapy, and less aggressive mechanical ventilation, this classic presentation of BPD has become relatively uncommon. At the same time, there has been a striking increase in survival rates among extremely premature infants, for whom underlying lung imma-turity plays a predominant role in the pathogenesis and clinical presentation of BPD. This new presentation has created confusions regarding the definitions and diagnostic criteria of BPD. Diagnostic challenges arise particularly with milder forms of BPD, where infants may have only mild initial respiratory failure and require shorter durations of respiratory support. In these cases, the radiographic findings can differ significantly from the classic pattern described by Northway and colleagues [10]. These various clinical and radiographic manifestations reflect the different pathogenic processes that underlines the new presentation of BPD.

Table 2. Results of echocardiography in patients with BPD.

| Echocardiography (n) | All 42 | Mild 16 | Moderate 14 | Severe 12 |
|---|---|---|---|---|
| Pulmonary hypertension (%) | 13 (30.9) | 6 (37.5) | 3 (21.4) | 4 (33.3) |
| Died | 2 (15.4) | 1 (16.7) | 0 | 1 (25) |
| Congenital heart disease % | 35 (83.3) | 12 (75) | 12 (85.7) | 11 (91.7) |
| Died | 9 (25.7) | 3 (25) | 2 (16.7) | 4 (36.4) |
| ASD (%) | 13 (37.1) | 4 (33.3) | 4 (33.3) | 5 (19.2) |
| VSD (%) | 5 (14.3) | 3 (25) | 1 (8.3) | 1 (3.8) |
| PDA (%) | 15 (42.9) | 4 (33.3) | 6 (50) | 5 (19.2) |
| PFO (%) | 2 (5.7) | 1 (8.3) | 1 (8.3) | 0 |
| At birth – 7 months | 28 | 7 | 11 | 10 |
| Pulmonary hypertension (%) | 9 (32.1) | 3 (42.9) | 3 (27.3) | 10 |
| Died | 2 (22.2) | 1 (33.3) | 0 | 1 (33.3) |
| Congenital heart disease % | 26 (92.9) | 7 (100) | 10 (90.9) | 9 (90) |
| ASD (%) | 10 (38.4) | 4 (57.1) | 3 (30) | 3 (33.3) |
| VSD (%) | 3 (11.5) | 1 (1.4) | 1 (10) | 1 (11.1) |
| PDA (%) | 12 (46.2) | 2 (2.9) | 5 (50) | 5 (55.6) |
| PFO (%) | 1 (3.8) | 0 | 1 (10) | 0 |
| After 7 months | 14 | 9 | 3 | 2 |
| Pulmonary hypertension (%) | 4 (28.6) | 2 (22.2) | 2 (66.7) | 0 |
| Died | 0 | 0 | 0 | 0 |
| Congenital heart disease % | 9 (64.3) | 5 (55.6) | 2 (66.7) | 2 (100) |
| ASD (%) | 3 (33.3) | 0 | 1 (50) | 2 (100) |
| VSD (%) | 2 (22.2) | 2 (40) | 0 | 0 |
| PDA (%) | 3 (33.3) | 2 (40) | 1 (50) | 0 |
| PFO (%) | 1 (11.1) | 1 (20) | 0 | 0 |

ASD-atrial septal defect; VSD-ventricle septal defect; PDA-patent ductus arteriosus; PFO-persistent foramen ovale.

Classic BPD was characterized by severe morphologic changes, including emphysema, atelectasis fibrosis, squamous metaplasia, and smooth muscle hypertrophy in the airways and pulmonary vasculature. These changes were associated with severe respiratory failure, airway obstruction, pulmonary hypertension, and cor pulmonale. In contrast, the milder forms of BPD that are more frequently observed today are mainly characterized by increased fluid, a diffuse inflammatory response, and significant decrease in alveolar septation and impaired vascular development [21–25]. These changes are more compatible with an arrest in lung development rather than mechanical injury. It remains unclear to what extent this developmental arrest is a result of premature lung exposure to gas breathing versus the effects of volutrauma and oxygen toxicity. Additional factors including incomplete development, inflammatory processes due to ante- or postnatal infections [26–30], and exposure of the immature pulmonary vasculature to increased flow through a persistent ductus arteriosus [31,32] are also implicated in the pathogenesis of BPD.

Limited data availability in our medical records constraints our assessment of long-term morbidity. However, neonatal sepsis (64.7%) and congenital heart disease (83.3%) emerged as prevalent comorbidities among infants with BPD, which atrial septal defect (ASD) and patent ductus arteriosus (PDA) being the most common congenital heart diseases. A retrospective study comparing BPD and respiratory distress syndrome (RDS) revealed a higher prevalence of PDA in BPD cases (64.5% vs 15.2%, $p < 0.0001$) [16], especially among severe BPD cases (95% vs 59%) [14]. Pulmonary hypertension is considered as a cardiovascular complication associated with BPD, although the mechanism responsible for elevated pulmonary vascular resistance and altered reactivity remains incompletely understood. Our study suggested

**Table 3. Multivariate analysis of the predictors of mortality in BPD infants before 7 months old.**

| Characteristic | Died n = 41 (%) | Survived N = 44 (%) | OR (CI 95%) | p | Adjusted OR (CI 95%) | P |
|---|---|---|---|---|---|---|
| Gender, male | 21 (51.2) | 24 (54.5) | 0.875 (0.373 −2.053) | 0.759 | | |
| Birth weight* | | | | | | |
| < 1000 g | 11 (26.8) | 9 (20.5) | 1.426 (0.521-3.903) | 0.490 | | |
| 1000 −< 1500 g | 18 (43.9) | 24 (54.4) | 0.652 (0.277-1.535) | 0.328 | | |
| 1500 −< 2500 g | 9 (22) | 7 (15.9) | 1.487 (0.497-4.445) | 0.478 | | |
| ≥ 2500 g | 3 (7.3) | 3 (6.8) | 1.079 (0.205-5.675) | 0.929 | | |
| Birth weight/ gestational age* | | | | | | |
| SGA | 12 (29.3) | 9 (20.5) | 1.609 (0.595-4.351) | 0.348 | | |
| AGA | 26 (63.4) | 34 (77.3) | 0.510 (0.197-1.317) | 0.164 | 0.360 (0.120-1.078) | 0.068 |
| LGA | 3 (7.3) | 1 (2.3) | 3.395 (0.339-34.025) | 0.299 | | |
| Surfactant therapy | 9 (22.0) | 7 (15.9) | 1.487(0.497-4.445) | 0.478 | | |
| Congenital anomaly | 3 (7.3) | 4 (9.1) | 1.267 (0.266-6.036) | 0.767 | | |
| Neonatal sepsis | 29 (70.7) | 26 (59.1) | 0.598 (0.243-1.473) | 0.263 | | |
| Oxygen supplementation | 34 (82.9) | 32 (72.7) | 0.549 (0.192-1.568) | 0.263 | | |
| Mechanical ventilation | 30 (73.2) | 22 (50) | 0.367 (0.148-0.910) | 0.031 | | |
| Respiratory problems (cough, dyspnea, wheezing) | 5 (12.2) | 2 (4.5) | 0.343 (0.063-1.875) | 0.217 | | |
| PMA< 28 week | 11 (26.8) | 4 (9.1) | 3.667 (1.063-12.651) | 0.040 | 0.253 (1.359–20.431) | 0.016 |
| Echocardiography (< 7 month) n = 28 | N = 8 (%) | N = 20 (%) | | | | |
| Pulmonary hypertension | 2 (25) | 7 (35) | 0.619 (0.098-3.919) | 0.610 | | |
| Congenital heart disease | 6 (75) | 20 (100) | 0.222 (0.036-1.390) | 0.108 | 0.44 (0.132-1.488) | 0.020 |

SGA-small for gestational age; AGA-appropriate for gestational age; LGA-large for gestational age; PMA-post-menstrual age; *each sub-class was compared to others sub-class to have 2x2 table.

a lower risk of PH in mild BPD cases compared to severe ones, although this association was not statistically significant (RR 0.72; CI95% 0.29–1.76).

The proportion of PH did not significantly differ among infants with mild, moderate and severe BPD. This finding was similar to previous study reported a PH prevalence as high as 25%, with a higher proportion in severe BPD (58%) than in moderate and mild cases [11]. Another study involving 42 preterm infants with BPD complicated by PH (gestational age < 32 weeks) reported that 43% had severe PH, with a mortality rate of 38% [33]. However, due to limitation in data availability, we could not calculate the rate of PH among preterm infants < 32 weeks of gestation with BPD in our study. Our findings revealed an overall mortality rate of 48.2% among infants with BPD, occuring within the first 7 month of age. Among BPD infants with PH, the mortality rate was 15.4%, with detection typically occuring between 2–4 months old. Otherwise, all BPD infants with PH detected before 2 months old survived, raising questions about whether early detection contributed to improved outcomes.

The lack of statistical significance in the incidence of pulmonary hypertension (PH) among patients with BPD may be attributed to incomplete ascertainment, as echo-cardiograms were not universally performed on all newborns with these comorbidities. McKinney et al. implemented a comprehensive screening protocol for PH and structural anomalies using echo-cardiography at 36 weeks PMA, with follow-up echo-cardiograms conducted before discharge or in response to significant clinical deterioration [34]. This systematic screening approach may have contributed to more complete detection of PH. Studies have shown that PH in BPD patients correlates with significantly lower cognitive, language, and motor scores on the Bayley III assessment at 18–24 months of age compared to non-PH infants, as well as lower z-scores for weight and head circumference [17]. These findings underscore the importance of close developmental monitoring and routine screening for cardiac issues in infants with BPD after discharge.

We observed a median length of stay of 57 days, with a trend toward longer hospitalizations associated with increasing severity of BPD. Interestingly, while mechanical ventilation was utilized more frequently in mild BPD cases (71.9%) compared to moderate (48.1%) and severe (61.5%), oxygen supplementation varied among the severity groups. Prolonged hospital stays may be attributed to the complexities of the clinical course, including factors such as sepsis and congenital heart disease, which often require extended stays in the neonatal intensive care unit (NICU). Further analysis is warranted to explore these associations. Identifying factors contributing to prolonged mechanical ventilation dependency may involve various modalities.for instance, bronchoscopy, a proven safe procedure in infants, allows dynamic assessment of airway issues [35], while chest computed tomography (CT) scans offer valuable insights into BPD severity and structural abnormalities detection [36].

The role of mechanical ventilation in bronchopulmonary dysplasia (BPD) is complex, balancing potential harm and necessary support. Initially, aggressive mechanical ventilation was linked to lung injury and severe respiratory symptoms in BPD due to volutrauma and barotrauma, which exacerbate inflammation and hinder lung development. However, our study found that mechanical ventilation was associated with a decreased risk of mortality among BPD infants under 7 months, suggesting its critical role in managing acute respiratory failure. This indicates that while mechanical ventilation can contribute to lung damage, it remains an essential intervention when used appropriately, particularly in severe cases of BPD. Therefore, a nuanced understanding of mechanical ventilation is necessary to optimize care strategies, maximizing its benefits while minimizing risks. Further research is needed to identify the best practices for ventilation that protect lung health in these vulnerable infants.

The higher use of mechanical ventilation in mild BPD patients compared to moderate or severe cases may stem from several factors. Mild BPD often presents with transient respiratory distress, prompting initial ventilation for stabilization. Hospital protocols might advocate for aggressive management of any respiratory issues, and variability in symptom presentation could lead to more cautious approaches. Additionally, surfactant therapy may influence the decision to initiate ventilation. Early intervention strategies and prolonged hospital stays for monitoring could also contribute to the increased use of mechanical ventilation in these infants. Overall, these factors reflect clinical practices aimed at preventing worsening respiratory conditions.

The mortality rate of BPD observed in our study was alarmingly high at 48.2%, with similar proportions observed across mild, moderate, and severe cases. This high mortality is not only attributable to the severity of BPD, but also to perinatal comorbidities such as neonatal sepsis and congenital heart disease. The majority of affected infants were premature (82.9%), with 66% born at less than 32 weeks' gestation and 92.7% with low birth weight, including 70.7% weighing less than 1500 g. Mortality predominantly occurred within the first 7 months after birth. Risk factors for mortality included a post-menstrual age of less than 28 weeks, while being appropriate for gestational age and the use of mechanical ventilation were identified as protective factors. A study in Korea reported improved survival rates among infants born at 23–26 weeks' gestation during 2006–2010 (80.3%) compared 2000–2006 (70.0%), especially in infants at 23–24 weeks' gestation (73.9% vs 47.4%, respectively). Improved survival was significantly associated with larger birth weight, higher Apgar score at 5 minutes, and maternal use of antenatal steroids before delivery [37]. The association between lower gestational age (GA) at delivery and mortality from BPD remains controversial. A gestational age of ≤ 28 weeks results in extreme structural and biochemical lung immaturity, which is the most powerful risk factor for the development of BPD. The increased survival of extremely premature infants might increase the actual number of premature infants at risk for BPD. However, Kim *et al*., found that increased survival of extremely preterm infants at 23–24 weeks gestation was associated with a reduced incidence of BPD in infants born at 25–26 weeks. Conversely, Botet *at al*., reported no increase in the survival among extremely low birth weight infants; although survival rate of infants without BPD improved from 58.5% (1997–2000) to 75% (2006–2009) [38]. Ambalavanan *et al*., employed statistical methods (Classification and Regression Tree/ CART) to identify risk factors for death and BPD, finding that increased severity of respiratory failure, lower birth weight, male gender and out-born status were major factors [39]. Intubation and ventilation may induce volutrauma and barotrauma, while less invasive methods such as continuous positive airway pressure (CPAP) could allow continuous alveolar

growth and reduce lung damage [40]. Notably, our study found mechanical ventilation to be a protective factor against mortality in infants with BPD, although the specific type of mechanical ventilation used were not detailed in our data. Further exploration and classification of ventilation modalities are necessary to elucidate their impact on patient outcomes.

The study adds to current knowledge by highlighting high mortality rates in infants with BPD in Yogyakarta, Indonesia, underscoring the critical role of low post-menstrual age (<28 weeks) as an independent predictor of mortality. Interestingly, mechanical ventilation was associated with lower mortality, contrasting with earlier findings where it was linked to lung damage in BPD. This suggests that careful, possibly less aggressive, ventilation strategies may improve survival. These results reinforce the importance of early, targeted interventions for high-risk infants in similar healthcare settings.

Our study has several limitations. Firstly, its retrospective nature relied on the accuracy of medical records, which may have led to incomplete data capture. Additionally, the study was conducted at a single hospital with a relatively small sample size, and approximately 15% of medical records were inactive, potentially introducing selection bias. Secondly, varying criteria for defining BPD over time highlight the need for standardized diagnostic criteria to ensure uniformity in diagnosis and data interpretation. Longitudinal studies involving multicenter and employing prospective cohort designs are warranted to provide a more comprehensive understanding of the long-term impact of BPD and chronic lung disease.

## Conclusion

In conclusion, our study a 1.2% incidence of BPD among hospitalized newborns, with a mortality rate affecting nearly half. Key predictors included neonatal sepsis, congenital heart disease, and a post-menstrual age of < 28 weeks, while mechanical ventilation appeared to reduce mortality. These findings underscore the importance of early identification and targeted management strategies to mitigate risks and improve outcomes for infants with BPD.

## Supporting information

**S1 File. Dataset BPD.**
(PDF)

## Acknowledgments

We gratefully acknowledge Melita Permata Sari, MD and Mohammad Buston, MD for providing the data collection and statistical support. We also thank to Nadya Arafuri, MD and Hendra Purnasida Bagaswoto, MD for helping data collection.

## Author contributions

**Conceptualization:** Dwi Kisworo Setyowireni, Roni Naning, Madarina Julia.

**Data curation:** Dwi Kisworo Setyowireni.

**Formal analysis:** Rina Triasih.

**Funding acquisition:** Dwi Kisworo Setyowireni.

**Investigation:** Dwi Kisworo Setyowireni.

**Methodology:** Roni Naning, Rina Triasih.

**Writing – original draft:** Dwi Kisworo Setyowireni.

**Writing – review & editing:** Indah Kartika Murni, Madarina Julia, Ekawaty Luthfia Haksari.

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
