## [Decision Letter · Decision Letter 0]

20 Sep 2023

Dear Dr. Dr. Setyowireni,

Thank you for submitting your manuscript to PLOS ONE. After careful consideration, we feel that it has merit but does not fully meet PLOS ONE’s publication criteria as it currently stands. Therefore, we invite you to submit a revised version of the manuscript that addresses the points raised during the review process.

I have read your report with interest but share the concerns of the reviewers. To make this paper ready for publication you will need to work on a major revision.

The reviewer comments will be a guide for your work.

The following is a summary of my comments: The outcomes reported are concerning and should be compared to most recent reports by others. More information is needed regarding risk factors and the reduction thereof.

Numbers regarding in-born versus out-born babies, extend of prenatal care, prenatal antibiotics, prenatal steroids, mode of delivery, description of Surfactant usage, available

Please submit your revised manuscript by Nov 04 2023 11:59PM. If you will need more time than this to complete your revisions, please reply to this message or contact the journal office at plosone@plos.org . A rebuttal letter that responds to each point raised by the academic editor and reviewer(s). You should upload this letter as a separate file labeled 'Response to Reviewers'.A marked-up copy of your manuscript that highlights changes made to the original version. You should upload this as a separate file labeled 'Revised Manuscript with Track Changes'.An unmarked version of your revised paper without tracked changes. You should upload this as a separate file labeled 'Manuscript'.

We look forward to receiving your revised manuscript.

Kind regards,

Barbara Wilson Engelhardt, MD

Academic Editor

PLOS ONE

Journal Requirements:

- https://doi.org/10.3346/jkms.2016.31.3.423

- https://doi.org/10.1053/j.semperi.2006.05.002

- http://dx.doi.org/10.1513/AnnalsATS.201709-756FR

- https://doi.org/10.3346/jkms.2016.31.3.423

In your revision ensure you cite all your sources (including your own works), and quote or rephrase any duplicated text outside the methods section. Further consideration is dependent on these concerns being addressed

Upon re-submitting your revised manuscript, please upload your study’s minimal underlying data set as either Supporting Information files or to a stable, public repository and include the relevant URLs, DOIs, or accession numbers within your revised cover letter. For a list of acceptable repositories, please see http://journals.plos.org/plosone/s/data-availability#loc-recommended-repositories . Any potentially identifying patient information must be fully anonymized.

Important: If there are ethical or legal restrictions to sharing your data publicly, please explain these restrictions in detail. Please see our guidelines for more information on what we consider unacceptable restrictions to publicly sharing data: http://journals.plos.org/plosone/s/data-availability#loc-unacceptable-data-access-restrictions . Note that it is not acceptable for the authors to be the sole named individuals responsible for ensuring data access.

4. PLOS requires an ORCID iD for the corresponding author in Editorial Manager on papers submitted after December 6th, 2016. Please ensure that you have an ORCID iD and that it is validated in Editorial Manager. To do this, go to ‘Update my Information’ (in the upper left-hand corner of the main menu), and click on the Fetch/Validate link next to the ORCID field. This will take you to the ORCID site and allow you to create a new iD or authenticate a pre-existing iD in Editorial Manager. Please see the following video for instructions on linking an ORCID iD to your Editorial Manager account: https://www.youtube.com/watch?v=_xcclfuvtxQ.

Reviewers' comments:

Reviewer's Responses to Questions

**Comments to the Author**

1. Is the manuscript technically sound, and do the data support the conclusions?

Reviewer #1: Yes

Reviewer #2: Partly

2. Has the statistical analysis been performed appropriately and rigorously?

Reviewer #1: Yes

Reviewer #2: N/A

3. Have the authors made all data underlying the findings in their manuscript fully available?

Reviewer #1: Yes

Reviewer #2: Yes

4. Is the manuscript presented in an intelligible fashion and written in standard English?

Reviewer #1: Yes

Reviewer #2: Yes

Reviewer #1: An interesting read. There are quite a few grammatical and sentence formulation errors which I think the authors should consider revising. Overall, these are my comments:

1) At the start of the discussion part, it has been mentioned that aggressive mechanical ventilation might be the cause of lung damage and severe respiratory symptoms in BPD. However, later, it has been said that mechanical ventilation had a mortality benefit in BPD patients. I believe the authors should clarify this area further to explain the role of mechanical ventilation in the disease. Devote a separate paragraph to summarise all the relevant findings in present study, existing literature and plausible explanation for the same.

2) In continuation with the above, what may be the reason that mild BPD patients had more frequent use of mechanical ventilation than moderate or severe BPD patients? The number of ventilator-days also seem to be more in mild BPD patients.

3) Considering the confusing findings and retrospective design of the study, the authors may do a power analysis for their primary outcome parameters to put things in perspective.

4) Finally, not just mortality but long-term morbidity of these subset of patients needs to be studied in future.

Reviewer #2: Dear Dwi Kisworo Setyowireni,

Thank you for submitting your manuscript entitled "Outcome of Infants with Bronchopulmonary Dysplasia" for consideration. I appreciate the opportunity to review your work, contributing to the ongoing discussion on bronchopulmonary dysplasia. After a careful evaluation, I have identified several areas that would benefit from further clarification and revision to enhance the manuscript's scientific rigor and readability. Below are my comments, broken down by section, which I hope you find constructive for refining your paper.

Introduction

1. Relevance of Citations: The introduction could benefit from more recent literature to establish the field's current state.

2. Problem Statement: The problem statement could be more precisely articulated. What is the specific gap in the existing literature that this study aims to fill?

3. Hypothesis/Research Question: It would be beneficial to clearly state the research questions or hypotheses. This would help the reader understand what to expect from the study.

Method

1. Sample Size Justification: There needs to be more discussion about how the sample size was determined. Was a power analysis conducted?

2. Demographics: More details about the participant demographics might make the sample more transparent.

3. Variables: It would be helpful to define the independent and dependent variables more precisely.

4. Statistical Tests: Mention the specific statistical tests that were employed for data analysis.

5. Ethical Considerations: No mention of ethical approval for human/animal studies. Is it applicable? If yes, it should be included.

Results

1. Data Presentation: More tables and/or graphs could be useful for visualizing the data.

2. Missing Data: There needs to be a mention of how missing data were handled.

3. Statistical Significance: Were the results statistically significant? P-values and confidence intervals could offer more insight.

Discussion

1. Interpretation of Findings: The discussion would benefit from a more in-depth interpretation. How do these results add to the current body of knowledge?

2. Limitations: The discussion of limitations is brief and could be more thorough.

3. Future Work: Suggestions for future research still need to be included. Addressing this could round out the paper effectively.

4. Practical Implications: Mention the practical implications of your findings could strengthen the impact of your paper.

Overall Comments

1. Cohesion and Flow: Ensure a logical flow of ideas from the introduction to the discussion.

2. Technical Jargon: Consider defining technical terms so the paper is accessible to a broader audience.

3. References: A uniform citation style is needed; some references are not in the same format as others.

4. Abstract: An abstract summarizing the entire paper, including key findings and implications, should be added.

5. Conclusions: A separate conclusions section summarizing key findings and their implications could make the paper more impactful.

**Do you want your identity to be public for this peer review?** For information about this choice, including consent withdrawal, please see our Privacy Policy

Reviewer #1: **Yes: ** Dr. Saikat Banerjee, MD Respiratory Medicine, Micromasters (MIT) in Statistics and Data Science

Reviewer #2: **Yes: ** Mohammad Arkani

---

## [Author Response · Author response to Decision Letter 1]

12 Nov 2024

Response to Reviewers' comments

Thank you very much for reviewing our paper and providing constructive and invaluable feedback. We have carefully considered your comments and have provided responses in blue font below each one.

Reviewer #1: An interesting read. There are quite a few grammatical and sentence formulation errors which I think the authors should consider revising. Overall, these are my comments:

Response:

We have carefully edited the grammatical and sentence errors and provide track changes and clean copy of manuscript.

1) At the start of the discussion part, it has been mentioned that aggressive mechanical ventilation might be the cause of lung damage and severe respiratory symptoms in BPD. However, later, it has been said that mechanical ventilation had a mortality benefit in BPD patients. I believe the authors should clarify this area further to explain the role of mechanical ventilation in the disease. Devote a separate paragraph to summarise all the relevant findings in present study, existing literature and plausible explanation for the same.

Response:

Thank you, we have added this paragraph into the discussion section:

The role of mechanical ventilation in bronchopulmonary dysplasia (BPD) is complex, balancing potential harm and necessary support. Initially, aggressive mechanical ventilation was linked to lung injury and severe respiratory symptoms in BPD due to volutrauma and barotrauma, which exacerbate inflammation and hinder lung development. However, our study found that mechanical ventilation was associated with a decreased risk of mortality among BPD infants under 7 months, suggesting its critical role in managing acute respiratory failure. This indicates that while mechanical ventilation can contribute to lung damage, it remains an essential intervention when used appropriately, particularly in severe cases of BPD. Therefore, a nuanced understanding of mechanical ventilation is necessary to optimize care strategies, maximizing its benefits while minimizing risks. Further research is needed to identify the best practices for ventilation that protect lung health in these vulnerable infants.

2) In continuation with the above, what may be the reason that mild BPD patients had more frequent use of mechanical ventilation than moderate or severe BPD patients? The number of ventilator-days also seem to be more in mild BPD patients.

Response:

Thank you, we have added this paragraph into the discussion section:

The higher use of mechanical ventilation in mild BPD patients compared to moderate or severe cases may stem from several factors. Mild BPD often presents with transient respiratory distress, prompting initial ventilation for stabilization. Hospital protocols might advocate for aggressive management of any respiratory issues, and variability in symptom presentation could lead to more cautious approaches. Additionally, surfactant therapy may influence the decision to initiate ventilation. Early intervention strategies and prolonged hospital stays for monitoring could also contribute to the increased use of mechanical ventilation in these infants. Overall, these factors reflect clinical practices aimed at preventing worsening respiratory conditions.

3) Considering the confusing findings and retrospective design of the study, the authors may do a power analysis for their primary outcome parameters to put things in perspective.

Response:

Thank you for the feedback. Yes, a power analysis was conducted and its value is 80.2%.

4) Finally, not just mortality but long-term morbidity of these subset of patients needs to be studied in future.

Response:

Thank you for your positive feedback. We indeed aim to extend our study to evaluate long-term morbidities of children with BPD and identify potential predictors. We have incorporated this additional sentence into the manuscript:

Longitudinal studies involving multiple institutions and employing prospective cohort designs are warranted to provide a more comprehensive understanding of the long-term impact and morbidities of BPD and chronic lung disease.

Reviewer #2: Dear Dwi Kisworo Setyowireni,

Thank you for submitting your manuscript entitled "Outcome of Infants with Bronchopulmonary Dysplasia" for consideration. I appreciate the opportunity to review your work, contributing to the ongoing discussion on bronchopulmonary dysplasia. After a careful evaluation, I have identified several areas that would benefit from further clarification and revision to enhance the manuscript's scientific rigor and readability. Below are my comments, broken down by section, which I hope you find constructive for refining your paper.

Response:

We sincerely appreciate your positive comments and the constrictive feedback you have provided. Thank you for your valuable insights and contributions to our work.

Introduction

1. Relevance of Citations: The introduction could benefit from more recent literature to establish the field's current state.

Response:

Thank you, we have added two recent literatures in the introduction section.

The long-term impact of BPD includes persistent respiratory symptoms and airflow obstruction, potentially affecting lung function into adulthood [2,3].

Reff 2. Tracy MK, Berkelhamer, SK. Bronchopulmonary Dysplasia and Pulmonary Outcomes of Prematurity. Pediatr Ann. 2019;48(4):e148-53.

Reff 3. Collaco JM, McGrath-Morrow SA. Bronchopulmonary Dysplasia as a Determinant of Respiratory Outcomes in Adult Life. Pediatr Pulmonol. 2021; 56(11): 3464–71. doi:10.1002/ppul.25301.

2. Problem Statement: The problem statement could be more precisely articulated. What is the specific gap in the existing literature that this study aims to fill?

Response:

Thank you for the feedback. We have revised the paragraph to clearly articulate the problem statement and gaps in the Introduction section below:

Despite the significant global burden of bronchopulmonary dysplasia (BPD), there is a notable scarcity of data regarding its incidence and outcomes in Indonesia. Therefore, this study aimed to evaluate the outcomes and predictors of mortality among children with BPD in Yogyakarta, Indonesia. By elucidating the management and prognosis of BPD in resource-limited settings, our findings aim to provide valuable insights into optimizing care strategies for affected children in similar contexts.

3. Hypothesis/Research Question: It would be beneficial to clearly state the research questions or hypotheses. This would help the reader understand what to expect from the study.

Response:

Thank you, we have incorporated additional sentence regarding the hypothesis below:

Our hypotheses were that the BPD burden is significant and simple predictors may independently predict mortality among Indonesian children with BPD.

Method

1. Sample Size Justification: There needs to be more discussion about how the sample size was determined. Was a power analysis conducted?

Response:

Thank you for the feedback. Sample size was calculated using formula for non-paired categorical comparative analysis using reference data from previous studies. A power analysis was conducted and its value was 80.2%.

2. Demographics: More details about the participant demographics might make the sample more transparent.

Response:

Thank you. This study included 65 subjects from Yogyakarta, 16 subjects from Central Java, 2 subjects from East Java, 1 subject from Papua, and 1 subject from West Kalimantan.

3. Variables: It would be helpful to define the independent and dependent variables more precisely.

Response:

Thank you, we have included these variables in the statistical analysis to elucidate their association with the dependent variable, mortality. The independent variables (predictors) included gender, birth weight, birth weight/gestational age, surfactant therapy, congenital anomaly, neonatal sepsis, mechanical ventilation, respiratory problems (cough, dyspnea, wheezing), PMA < 28 weeks, the presence of pulmonary hypertension and congenital heart disease.

4. Statistical Tests: Mention the specific statistical tests that were employed for data analysis.

Response:

A multivariate logistic regression model was used to identify independent predictors of mortality in infants with BPD.

5. Ethical Considerations: No mention of ethical approval for human/animal studies. Is it applicable? If yes, it should be included.

Response:

This study has been approved by Ethical Committee Faculty of Medicine, Public Health and Nursing Universitas Gadjah Mada (KE/FK/0171/EC/2021).

Results

1. Data Presentation: More tables and/or graphs could be useful for visualizing the data.

Response:

Thank you, we have provided three tables and a figure to visualize the data.

Table 1. Characteristics of 85 BPD patients in Yogyakarta, Indonesia.

Table 2. Echocardiographic findings in patients with BPD

Table 3. Multivariate analysis of the predictors of mortality in BPD infants before the age of 7 months old

Figure 1. Patient selection algorithm

2. Missing Data: There needs to be a mention of how missing data were handled.

Response:

Thank you, we addressed missing data using complete case Analysis. This approach involved analysing only those cases where data were available for all variables of interest. We have added this into the Method section.

3. Statistical Significance: Were the results statistically significant? P-values and confidence intervals could offer more insight.

Response:

Thank you, we have provided both p values and 95% confidence intervals in the manuscript.

Discussion

1. Interpretation of Findings: The discussion would benefit from a more in-depth interpretation. How do these results add to the current body of knowledge?

Response:

Thank you, we have added a paragraph to address this in the discussion session:

The study adds to current knowledge by highlighting high mortality rates in infants with BPD in Yogyakarta, Indonesia, underscoring the critical role of low post-menstrual age (<28 weeks) as an independent predictor of mortality. Interestingly, mechanical ventilation was associated with lower mortality, contrasting with earlier findings where it was linked to lung damage in BPD. This suggests that careful, possibly less aggressive, ventilation strategies may improve survival. These results reinforce the importance of early, targeted interventions for high-risk infants in similar healthcare settings.

2. Limitations: The discussion of limitations is brief and could be more thorough.

Response:

Thank you, we have elaborated the limitations of our study into this paragraph:

Our study has several limitations. Firstly, its retrospective nature relied on medical records, which may have led to incomplete data capture. Additionally, the study was conducted at a single hospital with a relatively small sample size, and approximately 15% of medical records were inactive, potentially introducing selection bias. Secondly, the criteria used to define BPD varied over time, reflecting evolving understanding of the condition. This inconsistency highlights the need for standardized criteria to ensure uniformity in diagnosis and data interpretation.

3. Future Work: Suggestions for future research still need to be included. Addressing this could round out the paper effectively.

Response:

Thank you, we have added this sentence for future study:

Longitudinal studies involving multiple medical institutions and employing prospective cohort designs are warranted to provide a more comprehensive understanding of the long-term impact and morbidities of BPD and chronic lung disease.

4. Practical Implications: Mention the practical implications of your findings could strengthen the impact of your paper.

Response:

Thank you, we have added a paragraph revealing the impact of our study to clinical practice below:

Our study revealed a significant burden of bronchopulmonary dysplasia, with nearly half of the infants diagnosed with BPD in our hospital died within seven months of life. Notably, post-menstrual age of less than 28 weeks was associated with an elevated risk of mortality in infants with BPD. These findings underscore the importance of informing policymakers to develop effective prevention strategies for BPD in Indonesian infants. Specifically, initiatives targeting the prevention and appropriate management of prematurity, particularly in infants with PMA less than 28 weeks, are important to mitigating mortality associated with BPD.

Overall Comments

1. Cohesion and Flow: Ensure a logical flow of ideas from the introduction to the discussion.

Response:

Thank you, we have made efforts to enhance the manuscript's cohesion and flowing.

2. Technical Jargon: Consider defining technical terms so the paper is accessible to a broader audience.

Response:

Thank you, we have revised the manuscript accordingly as suggested.

3. References: A uniform citation style is needed; some references are not in the same format as others.

Response:

Thank you, we have revised the manuscript accordingly as suggested.

4. Abstract: An abstract summarizing the entire paper, including key findings and implications, should be added.

Response:

Thank you, we have revised the manuscript accordingly as suggested.

5. Conclusions: A separate conclusions section summarizing key findings and their implications could make the paper more impactful.

Response:

Thank you, we have revised the manuscript accordingly as suggested.

---

## [Decision Letter · Decision Letter 1]

5 Nov 2025

Outcome of Infants with Bronchopulmonary Dysplasia

PONE-D-23-09921R1

Dear Dr. Setyowireni,

We’re pleased to inform you that your manuscript has been judged scientifically suitable for publication and will be formally accepted for publication once it meets all outstanding technical requirements.

Kind regards,

Stefan Grosek, Ph.D., M.D.,

Academic Editor

PLOS ONE

Additional Editor Comments (optional):

Reviewers' comments:

Reviewer's Responses to Questions

**Comments to the Author**

Reviewer #1: All comments have been addressed

Reviewer #3: (No Response)

Reviewer #4: All comments have been addressed

2. Is the manuscript technically sound, and do the data support the conclusions?

Reviewer #1: (No Response)

Reviewer #3: No

Reviewer #4: Yes

3. Has the statistical analysis been performed appropriately and rigorously?

Reviewer #1: (No Response)

Reviewer #3: No

Reviewer #4: Yes

4. Have the authors made all data underlying the findings in their manuscript fully available?

Reviewer #1: (No Response)

Reviewer #3: Yes

Reviewer #4: Yes

5. Is the manuscript presented in an intelligible fashion and written in standard English?

Reviewer #1: (No Response)

Reviewer #3: No

Reviewer #4: Yes

Reviewer #1: (No Response)

Reviewer #3: (No Response)

Reviewer #4: The article is very interesting, with a clear goal, well done with conclusions and relevant, adequate literature.

Acceptable for publication based on review

**Do you want your identity to be public for this peer review?** For information about this choice, including consent withdrawal, please see our Privacy Policy

Reviewer #1: **Yes: ** Dr. Saikat Banerjee, MD, DM, Pulmonary, Critical Care and Sleep Medicine, MITx Micromasters in Statistics and Data Science

Reviewer #3: No

Reviewer #4: **Yes: ** Miljana Z Jovandaric

---

## [Editor Report · Acceptance letter]

PONE-D-23-09921R1

PLOS ONE

Dear Dr. Setyowireni,

I'm pleased to inform you that your manuscript has been deemed suitable for publication in PLOS ONE. Congratulations! Your manuscript is now being handed over to our production team.

Kind regards,

on behalf of

Professor Stefan Grosek

Academic Editor

PLOS ONE